# Generation of Spike-Extracellular Vesicles (S-EVs) as a Tool to Mimic SARS-CoV-2 Interaction with Host Cells

**DOI:** 10.3390/cells11010146

**Published:** 2022-01-03

**Authors:** Roberta Verta, Cristina Grange, Renata Skovronova, Adele Tanzi, Licia Peruzzi, Maria Chiara Deregibus, Giovanni Camussi, Benedetta Bussolati

**Affiliations:** 1Department of Molecular Biotechnology and Health Sciences, University of Turin, 10126 Turin, Italy; roberta.verta@unito.it (R.V.); renata.skovronova@unito.it (R.S.); adele.tanzi@unito.it (A.T.); 2Department of Medical Science, University of Turin, 10126 Turin, Italy; cristina.grange@unito.it (C.G.); giovanni.camussi@unito.it (G.C.); 3Pediatric Nephrology Unit, Regina Margherita Children’s Hospital, Città della Salute e della Scienza di Torino, 10126 Turin, Italy; licia.peruzzi@unito.it; 42i3T Business Incubator and Technology Transfer, University of Turin, 10126 Turin, Italy; mariachiara.deregibus@unito.it

**Keywords:** extracellular vesicles, COVID-19, SARS-CoV-2, SARS-CoV-2 spike protein, colchicine, anti-ACE2

## Abstract

Extracellular vesicles (EVs) and viruses share common features: size, structure, biogenesis and uptake. In order to generate EVs expressing the SARS-CoV-2 spike protein on their surface (S-EVs), we collected EVs from SARS-CoV-2 spike expressing human embryonic kidney (HEK-293T) cells by stable transfection with a vector coding for the S1 and S2 subunits. S-EVs were characterized using nanoparticle tracking analysis, ExoView and super-resolution microscopy. We obtained a population of EVs of 50 to 200 nm in size. Spike expressing EVs represented around 40% of the total EV population and co-expressed spike protein with tetraspanins on the surfaces of EVs. We subsequently used ACE2-positive endothelial and bronchial epithelial cells for assessing the internalization of labeled S-EVs using a cytofluorimetric analysis. Internalization of S-EVs was higher than that of control EVs from non-transfected cells. Moreover, S-EV uptake was significantly decreased by anti-ACE2 antibody pre-treatment. Furthermore, colchicine, a drug currently used in clinical trials, significantly reduced S-EV entry into the cells. S-EVs represent a simple, safe, and scalable model to study host-virus interactions and the mechanisms of novel therapeutic drugs.

## 1. Introduction

The outbreak of severe acute respiratory syndrome β-coronavirus 2 (SARS-CoV-2), causing coronavirus disease 2019 (COVID-19), represents the current health concern around the world. Coronaviruses (CoVs) are enveloped viruses with positive-sense 5′-3′ single-stranded RNA of the Coronaviridae family [1]. These viruses are around 125 nm particles and contain a viral genome of around 30 (26–32) kb pairs. The virions have a structural spike glycoprotein, an M-membrane protein (a type III transmembrane glycoprotein), an N-nucleocapsid protein (which is present within the phospholipid bilayer), and non-structural proteins [1]. The SARS-CoV infection begins with the virus binding to the angiotensin-converting enzyme2 (ACE2) [2]. ACE2 is widely expressed in human, including lung alveolar epithelial cells, small intestinal epithelial cells, cardiovascular system, central nervous system, and kidney. These targets play an important role in COVID-19 pathophysiology [3,4]. The ACE2 binding and subsequent CoVs entry into host cells is mediated by the spike glycoprotein that is composed of two functional subunits, the S1 and S2 [2,5]. The S1 subunit consists of an N-terminal domain and a receptor binding domain and acts to bind to the receptor of the host cell. The S2 subunit subsequently fuses the virus with cell membranes [2,5,6]. The spike protein is mainly cleaved by Furin, present on the host cell surface membrane, into the S1 and S2 components corresponding to the prefusion state. The subsequent fusion is considered to involve a second cleavage by a serine protease or by endosomal cysteine proteases, triggering S1 dissociation and irreversible S2 folding into a fusion state conformation. These major structural rearrangements are required for cell and viral membrane fusion and for the viral RNA release into the cytoplasm. Therefore, the spike glycoprotein is crucial for the entry of SARS-CoV-2 and represents an excellent target for anti-viral therapeutic development [2,5,6,7].

Extracellular vesicles (EVs) are cell-derived membranous vesicles present in biological fluids, important for cell-to-cell communication through the release of bioactive factors (proteins, lipids and genetic material) and are involved in physiological and pathological processes [8,9]. It is already known that EVs and viruses share common aspects: their biogenesis, uptake, and the ability to carry a specific cargo while being different entities [10,11]. Recent findings demonstrate that viruses take advantage of EVs for cellular release, and EVs control viral entry mechanisms for cargo delivery [10]. The viruses use EV endocytic routes to enter uninfected cells and change the EV secretory pathway to exit infected cells, thus illustrating that EVs and viruses share common cell entry and biogenesis mechanisms [11]. Furthermore, exosomes from infected cells contain viral components, which are important mediators of antiviral responses which make them ideal for a new vaccine, as well as vehicles that facilitate the spread of viral infection [12]. 

On the other side, the engineering of EVs could be interesting for anti-viral purposes. A recent study demonstrated that ACE2-engineered EVs limit the SARS-CoV-2 infection [13]. Another work showed the possibility of using EVs modified with the receptor-binding domain of the viral spike protein that recognizes ACE2 receptor as a target delivery system in vivo of potential anti-viral agents [14]. Recently, Troyer Z et al. showed that EVs containing SARS-CoV-2 spike interact with the humoral immune system and reduce serum neutralizing antibodies of convalescent patients [15].

Therefore, this study aimed to generate a simple, safe, and scalable model to study therapeutic approaches for the blocking of SARS-CoV-2 cell binding and entry using EV properties. In particular, we generated EVs that present the spike protein of SARS-CoV-2, essential for virus–cell fusion and entry into host cell following ACE2 receptor bind [3,4,16]. To obtain these modified EVs, we started from cells transfected with the spike vector, an indirect engineering method. We analyzed the spike-extracellular vesicles (S-EVs)–endothelial cells interaction, and we evaluated the effect of anti-ACE2 blocking antibody and colchicine, a drug under clinical trial for COVID-19 treatment [17].

## 2. Materials and Methods

### 2.1. Cell Cultures

Primary human umbilical vein endothelial cells (HUVEC) were purchased from ATCC (ATCC-PCS-100-010, Manassas, VA, USA) and cultured in the EndoGRO VEGF Supplement Kit (Millipore Sigma™, Burlington, MA, USA) adding 5% foetal bovine serum (FBS; Euroclone, Milan, Italy), and all experiments were performed between passages 2 and 5. The immortalized normal human bronchial epithelial (16HBE14o−) cell line was kindly provided by Dr. Alessandra Ghigo (University of Turin, Italy) who originally received the cells from Dr. Dieter Gruenert (University of California San Francisco, San Francisco, CA, USA). 16HBE14o- were cultured in Minimum Essential Medium (MEM) (Lonza, Basel, Switzerland) supplemented with 10% of FBS (Euroclone, Milan, Italy) and 100 U/mL of penicillin/streptomycin on culture dishes pre-coated with human fibronectin (1 mg/mL; Sigma-Aldrich, Saint Louis, MO, USA), bovine collagen I (3 mg/mL; Sigma-Aldrich, Saint Louis, MO, USA) and bovine serum albumin (0.1%), as previously described [18]. Human embryonic kidney cells (HEK293T) SARS-CoV-2 spike-transfected (H-S) or not transfected (H-C) cell lines were purchased from LiStarFish (Milan, Italy) and cultured in the high-glucose Dulbecco’s Modified Eagle Medium (DMEM) (Thermo Fisher Scientific, Waltham, MA, USA) with 100 U/mL of penicillin/streptomycin. The H-S were transfected with mammalian expression vector, the pCMV3-2019-nCoV-Spike (S1+S2)-long plasmid. The transfection quality control was confirmed by full-length sequencing using the primers pCMV3-F: 5′ CAGGTGTCCACTCCCAGGTCCAAG 3′, pcDNA3-R: 5′ GGCAACTAGAAGGCACAGTCGAGG 3′ or T7-F: 5′ TAATACGACTCACTATAGGG 3′, BGH-R: 5′ TAGAAGGCACAGTCGAGG 3′) and validated by the expression of SARS-CoV-2 spike protein in cells surface membrane. Hygromycin (80 µg/mL) was added to the H-S medium during every 3 passages to select the transfected cells. Human lung fibroblast cells (MRC5) were purchased from Sigma-Aldrich (St. Louis, MO, USA) and were cultured in DMEM low glucose in the presence of 10% FBS and 100 U/mL of penicillin/streptomycin. The cell culture incubation was performed in incubator at 37 °C with 5% CO_2_ and controlled humidity. 

### 2.2. EV Isolation and Characterization

The S-EVs and C-EVs were obtained from supernatants of H-S or H-C, respectively, cultured 16 h in RPMI deprived of FBS. After removal of cell debris and apoptotic bodies by centrifugation at 3000 *g* for 20 min, EVs were purified by 2 h ultracentrifugation at 100,000 *g* at 4 °C (Beckman Coulter Optima L-90 K; Fullerton, CA, USA). EVs were used fresh or stored at −80 °C after resuspension in RPMI supplemented with 1% dimethyl sulfoxide (DMSO). Analysis of size distribution and enumeration of EVs were performed using nanoparticle tracking analysis NS300 (Malvern Instruments Ltd, Malvern, UK) equipped with a 488 nm laser and the nanoparticle tracking analysis 3.2 software. 

#### 2.2.1. ExoView Analysis

Characterization of S-EVs markers was performed by using an ExoView Tetraspanins Kit (NanoView Bioscience, Boston, MA, USA). Each chip was coated with CD9, CD63, CD81 antibodies and MIgG control antibody. The chips were incubated with EV samples, using 35 μL of EV (1 × 10^9^ particles/mL according to the nanoparticle tracking analysis) suspension, left overnight and protected from the light. After multiple washing steps, the chips were analyzed using ExoView^TM^ R100 imaging platform (NanoView Bioscience, Bioscience, Boston, MA, USA) with ExoViewer software.

#### 2.2.2. Super-Resolution Microscopy

Super-resolution microscopy analyses of S-EVs were performed using a temperature-controlled Nanoimager S Mark II microscope from ONI (Oxford Nanoimaging, Oxford, UK) equipped with a 100×, 1.4NA oil immersion objective, an XYZ closed-loop piezo 736 stage, and 405 nm/150 mW, 473 nm/1 W, 560 nm/1 W, 640 nm/1 W lasers and dual/triple emission channels split at 640 and 555 nm. 

The samples were prepared using 10 μL of 0.01% Poly-L-Lysine (Sigma-Aldrich, St. Louis, MO, USA) placed on cleaned high-precision coverslips and were placed at 37 °C in a humid chamber for 2 h. After this time, excess Poly-L-Lysine was removed. A total of 1 μL of EVs (1 × 10^10^) resuspended in 9 μL of blocking solution (PBS-5% Bovine Serum Albumin) was pipetted into a previously coated well to attach overnight at +4 °C. The next day, the sample was removed, and 10 μL of blocking solution was added into the wells for 30 min. Then, 2.5 μg of purified mouse anti-CD9, anti-CD63, anti-CD81 (Oxford Nanoimaging, Oxford, UK) and anti-SARS-CoV-2 S2 antibody (LiStarFish, Milan, Italy) were conjugated with Alexa Fluor 555, 647, or 488 dyes using the Apex Antibody Labeling Kit (Invitrogen, Carlsbad, CA, USA), according to the manufacturer’s protocol. The antibodies were left for overnight incubation at +4 °C protected from the light. The samples were washed twice with PBS and a 10 µL ONI BCubed Imaging Buffer (Alfatest, Roma, Italy) was added for amplifying the EV fluorescence signal. Three-channel dSTORM data (2000 frames per channel) were acquired sequentially at 30Hz (Hertz) in the total reflection fluorescence (TIRF) mode. Single molecule data were filtered using NimOS (Version 1.18.3, ONI), based on the point spread function shape, photon count, and localization precision to minimize background noise and remove low-precision localizations.

All pictures were analyzed by the CODI website platform www.alto.codi.bio (ONI). The filtering and drift correction was used as in NimOS software. The BDScan clustering tool was applied to merged channels, and EVs were counted co-localized or in separate channels.

#### 2.2.3. Transmission Electron Microscopy

Transmission electron microscopy (TEM) was performed on S-EVs placed on 200-mesh nickel formvar carbon-coated grids (Electron Microscopy Science, Hatfield, PA, USA) and left to adhere for 20 min. The grids were then incubated with 2.5% glutaraldehyde containing 2% sucrose and, after washings in distilled water, the EVs were negatively stained with NanoVan (Nanoprobes, Yaphank, NY, USA) and observed using a Jeol JEM 1010 electron microscope (Jeol, Tokyo, Japan).

#### 2.2.4. MACSPlex Exosome Kit Analysis

Samples were subjected to bead-based multiplex EV analysis by flow cytometry (MACSPlex Exosome Kit, human, Miltenyi Biotec, CA, USA), 1 × 10^9^ EV containing samples (concentration normalized using nanoparticle tracking analysis) were processed as follows: samples were diluted with MACSPlex Buffer (MPB) to a final volume of 120 µL. 15 µL of MACSPlex Exosome Capture Beads (containing 39 different antibodies-coated bead subsets) were added to each sample. Samples were then incubated on an orbital shaker overnight (14–16 h) at 450 rpm at +4 °C, protected from light. To wash the beads, 1 mL of MPB was added and removed after several centrifugations (3000 *g*, 5 min). For counterstaining of EV bound by capture beads with detection antibodies, 135 µL of MPB and 5 µL of each APC-conjugated anti-CD9, anti-CD63, and anti-CD81 detection antibodies (provided in kit) were added to each sample and were incubated on an orbital shaker at 450 rpm protected from light for 1 h at room temperature. After that, 1mL of MPB was added to wash the beads and then it was removed after one centrifugation (3000 *g*, 5 min). This step was followed by another washing with 200 µL of MPB, incubation on an orbital shaker at 450 rpm protected from light for 15 min at room temperature and then MPB was removed. Subsequently, 150 µL of MPB was added to each sample and flow cytometric analysis was performed.

### 2.3. Uptake of DiI-labeled EVs in Target Cells 

In brief, 2.4 µL of vibrant^TM^ DiI cell-labeling solution (Invitrogen, Carlsbad, CA, USA) was added to EV samples and incubated for 20 min at 37 °C. After the incubation, the labeled-EVs were purified with 1h ultracentrifugation at 100,000 *g* at +4 °C and resuspended in RPMI +1% DMSO. A DiI control solution (CTL-DiI) was prepared using the protocol above, in the absence of EVs. HUVEC were incubated with 40,000 DiI-labeled EVs/target cells at 37 °C for different time points (30 min, 1 h, or 3 h) to monitor EV internalization over time. In selected experiments, HUVEC were treated with anti-ACE2 blocking antibody at the concentration of 20 µg/mL (AF933, R&D Systems, Minneapolis, MN, USA) or colchicine 150 µM for 2 h. After the treatments, DiI-labeled S/C-EVs were added to the medium for 3 h; at the end of experiments, cells were subjected to immunofluorescence analysis. Cells were extensively washed with PBS and fixed in paraformaldehyde (PAF) 4%. The FITC-phalloidin (Sigma-Aldrich, St. Louis, MO, USA) was used to label actin filaments of HUVEC and nuclei were stained with 4.6-diamidine-2-phenylindole (DAPI) (Sigma-Aldrich, St. Louis, MO, USA). Cells and EVs fluorescence were evaluated using an Apotome fluorescent microscope (Zeiss, Jena, Germany), magnification 40×.

### 2.4. Cytofluorimetric Analysis

HUVEC, 16HBE14o-, MRC5, H-S and H-C were detached using a nonenzymatic cell dissociation solution and resuspended in PBS 0.1% BSA (Sigma-Aldrich, St. Louis, MO, USA) and incubated with antibodies. The following antibodies conjugated with phycoerythrin (PE), fluorescein isothiocyanate (FITC) or allophycocyanin (APC), were used: CD54 (559771), CD146 (550315) from BD Bioscences (Franklin Lakes, NJ, USA), -CD29 (130-101-256), CD31 (130-092-652), CD105 (130-094-941), CD45 (170-081-063), VEGFR-2 (130-095-324) from Miltenyi Biotec (Bergisch Gladbach, Germany), ACE2 (AG-20A-0032TD) from Adipogen (Adipogen Life Sciences, CA, USA) and spike (CRE-CABT-CS048B) from LiStarFish (Milan, Italy) conjugated with Alexa Fluor 488 dyes using the Apex Antibody Labeling Kit, Invitrogen (Carlsbad, CA, USA). Moreover, DiI-labeled EV uptake by HUVEC and 16HBE14o- was evaluated, after the pre-treatments describe above, using cytofluorimetric analysis.

### 2.5. Western Blot

For protein analysis, the H-S and H-C and S/C-EVs were lysed at 4 °C for 30 min in RIPA buffer (20 nM Tris·HCl, 150 nM NaCl, 1% deoxycholate, 0.1% SDS 1% Triton X-100, pH 7.8) supplemented with protease and phosphatase inhibitor cocktail and PMSF (Sigma-Aldrich, St. Louis, MO, USA). Total protein concentration was determined spectrophotometrically using a micro-BCA™ Protein Assay Kit, as previously described [19]. Proteins were separated by 4% to 20% gradient sodium dodecyl sulfate–polyacrylamide gel electrophoresis (SDS PAGE, Biorad, Milan, Italy) and subjected to immunoblotting using the following primary antibodies: rabbit polyclonal anti-S2 (CRE-CABT-CS048B, LiStarFish, Milan, Italy), mouse monoclonal anti-CD63 (sc-5275, Santa Cruz Biotechnology, Heidelberg, Germany) used as a positive control for EVs, anti-Calreticulin (#2891 Cell Signaling Technology, Milan, Italy) used as negative control for EVs and mouse monoclonal anti-GAPDH (sc-47724, Santa Cruz Biotechnology, Heidelberg, Germany) was used as housekeeping for the cells. The protein bands were detected using rabbit or mouse peroxidase-labeled secondary antibody and visualized using an enhanced chemiluminescence detection kit and ChemiDoc™ XRS+ System (BioRad, Milan, Italy).

### 2.6. Statistical Analysis

Data are shown as mean ± SD. At least three independent replicates were performed for each experiment. Statistical analysis was carried out on Graph Pad Prism version 8.0.1 (GraphPad Software, Inc., San Diego, CA, USA) using the Paired *t*-test followed by ratio paired *t*-test and unpaired *t*-test followed by Mann Whitney test. Significance was set at probability value of *p* < 0.05.

## 3. Results

### 3.1. S-EV Generation and Characterization

We first validated the presence of spike on transfected cells by cytofluorimetric analysis (Figure 1A). Cells were analyzed every 3 passages with comparable spike expression (data not shown). Transfection did not alter cell phenotype, as displayed by maintenance of the typical expression of CD146 and CD29 progenitor markers [20,21,22,23] respect to the H-C (Figure 1B). We demonstrated the expression of the full-length spike protein and of the lower molecular weight S2 subunit, after S1 cleavage, in the H-S by western blot (Figure 1C). Afterwards, we isolated the S-EVs or C-EVs by 2 h ultracentrifugation at 100,000 *g* at +4 °C. EVs were used fresh or stored at −80 °C after resuspension in RPMI supplemented with 1% DMSO. The S-EVs were subjected to TEM analysis confirming their typical cup-shaped morphology and a size of about 100 nm (Figure 1D). We did not observe any differences in S-EVs and C-EVs concentration and size distribution by the nanoparticle tracking analysis (Figure 1E,F), obtaining homogenous population with a size between 50 to 200 nm (Figure 1E,F). Western blot analysis confirmed the presence of the full-length spike, and in low amount of the S2 subunit, in the EVs obtained from H-S but not from H-C (Figure 1G). 

Moreover, EVs were characterized by surface marker expressions, including tetraspanins, and typical markers of HEK293T cells, using a MACSPlex Exosome analysis kit after bead-based immunocapture. The S-EVs resulted positive for all exosomal markers and for some progenitor cell surface markers as the control EVs, indicating that transfection did not alter surface marker expression (Figure 2).

To better characterize spike-expressing EVs at a single EV level, EVs size, morphology and the co-localization of tetraspanins with spike protein were assessed using super-resolution microscopy and by ExoView chip-based analysis. Super-resolution microscopy confirmed spike expression by EVs, coupled with one, two, or three tetraspanins CD9, CD63, and CD81 (Figure 3). By CODI analysis, 19% of EVs were triple positive for the spike, CD63 and CD9 (Figure 3B) and 13% of EVs were triple positive for the spike, CD81 and CD9 (Figure 3C). A total of 11% and 4% were double positive for spike with CD63 or spike with CD81, respectively (Figure 3B,D). The spike and CD9 coexpression was between 6 and 11% (Figure 3B,D). The percentage of EVs positive only for the spike was constant at 7% of expression (Figure 3B,D). In total, S-EVs represented between 35 and 43% of the total tetraspanin expressing population (Figure 3B,D). Moreover, the super-resolution microscopy analysis confirmed the EV size assessed by TEM and nanoparticle tracking analysis resulting in a mean size of about 100 nm (Figure 3A,C).

The co-expression of spike with CD9, CD63 and CD81, on the EVs surface, was further confirmed using ExoView analysis, with similar expression levels on the single tetraspanin-affinity chips (Figure 4B). 

### 3.2. Uptake of S-EVs by HUVEC

Endothelial activation and dysfunction participate in COVID-19 pathogenesis by altering the integrity of the vessel barrier, promoting a pro-coagulative state, and inducing endothelial inflammation and leukocyte infiltration [24,25]. Therefore, we focused on the S-EV/HUVEC interaction. We characterized HUVEC for the expression of the typical endothelial markers (Appendix A), confirming the presence of CD31, CD105, CD146, CD54, VEGFR-2 endothelial markers and not of CD45 [26,27,28,29]. Moreover, we confirmed the ACE2 (spike receptor) expression by HUVEC, as demonstrated by previous studies [30,31,32] using cytofluorimetric analysis (Appendix A). Therefore, we analyzed the possible interaction of S-EVs with target cells. We demonstrated a time-dependent uptake of fluorescently labeled S-EVs or C-EVs by HUVEC. S-EVs were more internalized than C-EVs at each experimental time point considered, as detected by the immunofluorescence analysis (Figure 5A). The best time point chosen for EV uptake was 3 h. Comparing the uptake of S-EVs and C-EVs, after 3 h, we confirmed a significantly higher entrance of S-EVs with respect to the C-EVs by cytofluorimetric analysis (Figure 6A,B).

### 3.3. Modulation of S-EV Uptake

We subsequently analyzed the effect of colchicine and anti-ACE2 blocking antibody on EV uptake. Anti-ACE2 blockade significantly inhibited the S-EV uptake by HUVEC. The effect was specific for S-EVs, as C-EV entrance in the cell was not impaired (Figure 5B and Figure 6A,C). Colchicine, a microtubule antagonist that inhibits the tubulin polymerization [33,34], significantly reduced the S-EV internalization, with a trend of reduction also for the C-EV uptake (Figure 6A,C). In parallel, colchicine altered HUVEC shape, inducing the loss of adhesiveness between cells (Figure 5B and Appendix A). 

In addition, we analyzed the S-EV uptake by a bronchial epithelial cell line 16HBE14o-, SARS-CoV-2 virus target cells [35], after colchicine or anti-ACE2 blocking antibody treatments. High ACE2 expression was assessed using cytofluorimetric analysis (Appendix A). The S-EV internalization into 16HBE14o- cells was significantly higher than the one of C-EVs (Figure 7A,B). In addition, colchicine significantly decreased the S-EV and the C-EV uptake by bronchial epithelial cells. At variance, anti-ACE2 blocking antibody significantly reduced the S-EV entrance only (Figure 7A,C). Our results support the S-EV binding to endothelial and bronchial cells through an ACE2-dependent interaction, in the same manner as the SARS-CoV-2 virus.

## 4. Discussion

In the present study, we obtained spike engineered EVs from HEK293T cells transfected with the spike vector to study host–virus interactions, and we characterized the level of spike expression at a single EV level. Moreover, we showed that ACE2 is involved in S-EV entrance into HUVEC and 16HBE14o- cells. Finally, we showed that colchicine may affect S-EV uptake, providing a rationale for its anti-viral effect. 

ACE2 has been identified as the SARS-CoV-2 receptor, providing a critical link between infection and immunity, and inflammation and cardiovascular disease [3]. In this study, we obtained, with an indirect method, spike expressing EVs. Indeed, we deeply characterized at a single EV level the S-EVs showing variable co-expression of all three tetraspanins (CD9, CD63 and CD81) in about half of the isolated EVs, suggesting that the spike incorporated in a specific subset of EVs. This study parallels a recent study showing the neutralizing effect of spike carrying EVs obtained from HEK transfected cells [15]. The analysis of the obtained EGFP-bound spike EVs by nanoflow cytometry showed 85% of CD9 co-expression, whereas CD81 was mainly negative. This discrepancy could be due to the different sensitivity of the detection systems used. However, the authors showed that the spike protein was incorporated predominantly as the S2 subunit with rarer full-length glycoproteins [15]. At variance, in our study, we detected by western blot analysis mainly the full-length form of the spike protein.

Moreover, spike expressing EVs were shown to act as decoy targets for convalescent patient serum-derived Abs, reducing their effectiveness in blocking viral entry [15]. Therefore, it is conceivable that the generation of spike expressing EVs that are naturally released from infected cells may participate in the disease with different mechanisms of viral dissemination and immune system downregulation [15]. 

From a different perspective, there is an increasing number of clinical investigations to find therapeutic solutions, including the possible application of EVs engineering as therapeutic elements, or, as in the present study, as a safe and valuable tool for the scientific community to identify therapies against COVID19, combining properties of EVs and their common characteristics with viruses. We here observed a significantly increased S-EV uptake by HUVEC with respect to the C-EVs. This effect could be adequately explained by the spike presence on the surface of EVs and its interaction with ACE2 on cells. However, it is possible that other EV characteristics and additional mechanisms for entrance may be involved [36]. The spike could act as an additional factor for EV uptake, supporting its primary role in the virus–host interaction [25]. 

It is essential to underline that our study did not take advantage of using cells that overexpress ACE2, but tested the interaction between virus-like particles and the basal ACE2 receptor expression on the target cells. This supports and confirms that the SARS-CoV-2 infectivity is strongly connected to the high affinity of the virus ACE2 receptor [3,4,37]. It was shown that the SARS-CoV-2 interface with HUVEC could induce an exacerbated endothelial dysfunction [24,25], increasing the risk of mortality, especially for people with pre-existing health conditions including diabetes, obesity, and pulmonary and cardiovascular disease [25,38]. Furthermore, the ACE2 receptor is expressed in airway epithelium that corresponds to the first site of virus infections [35,39,40]. Therefore, we analyzed the S-EV interaction with normal human bronchial epithelial cells. We demonstrated that the S-EV internalization is dependent on ACE2 interaction, as for the SARS-CoV-2 virus. In the presence of ACE2-blocking antibody, we observed a statistically significant reduction of S-EV uptake by the HUVEC, but not in the case of C-EVs. To corroborate the findings obtained by EV uptake verified with HUVEC, we used bronchial epithelial cells, the main site of SARS-CoV-2 infection [35,39,40]. We confirmed the reduction of S-EV internalization by anti-ACE2-blocking antibody also using these cells. As S-EVs mimic SARS-CoV-2 interaction with host cells, it appears as an important resource in this scenario in identifying new therapeutic strategies.

Indeed, EVs have been considered a potential strategy to block ACE2, by competitive binding to neutralize the virus and to prevent the virus–host cell interaction. ACE2 expressing mesenchymal stem cells derived EVs appeared able to bind the SARS-CoV-2 competitively and were proposed as a possible COVID-19 therapy [41]. Moreover, EVs containing ACE2, alone or in combination with transmembrane protease serine 2 (TMPRSS2), from transfected 293FT mock cell, block SARS-CoV-2 spike-dependent infection in a much more efficient manner than soluble ACE2 [13]. 

Recently, colchicine has emerged as a therapy of interest for the treatment of COVID-19. Colchicine is a known drug widely used in autoimmune and inflammatory disorders [42], which works by inhibiting the polymerization of microtubules, a key component of the cell cytoskeleton [33,34]. The rationale for the use of colchicine in COVID-19 is based on its well-known anti-inflammatory, anti-fibrotic properties and its theoretical antiviral action, indirectly supported by the role of microtubules for the entry of the human coronavirus [42,43]. This interference with microtubule polymerization influences the macrophage diapedesis, endocytosis, and exocytosis, and consequently the interleukins (ILs) production [44,45,46]. Recently, it has been shown how colchicine inhibits the NOD-like receptor family pyrin domain containing 3 (NLRP3) inflammasome, possibly through its microtubule antagonism [47,48] and therefore blocks the IL-1 and IL-18 formation [49,50,51]. In addition, colchicine showed an impressively rapid effect on endothelial hyper-permeability observed in the capillary leak syndrome [52]. In the present study, we noted not only a significant reduction in S-EV uptake by HUVEC and by 16HBE14o- with colchicine treatment, but also its direct effect on the cytoskeleton and shape of the cells, as observed in other studies [45,53]. Interestingly, colchicine treatment was able to prevent S-EV entry with a stronger effect than that of ACE2 neutralizing antibody. Moreover, it also prevented the entry of C-EVs into cells, in particular into bronchial epithelial cells, suggesting an additional effect due to activity on microtubules and cell cytoskeleton. This aspect could be of interest for further studies aimed at blocking EV entry in pathologies involving EV-mediated spread of the disease.

## 5. Conclusions

The development of safe and effective vaccines, therapeutics, and drug delivery systems to the target site is a field that has increasingly gained attention to overcome SARS-CoV-2. This work demonstrates the possible use of S-EVs as a safe method for the study of COVID-19 and for the development of new therapeutic strategies.

## Figures and Tables

**Figure 1 cells-11-00146-f001:**
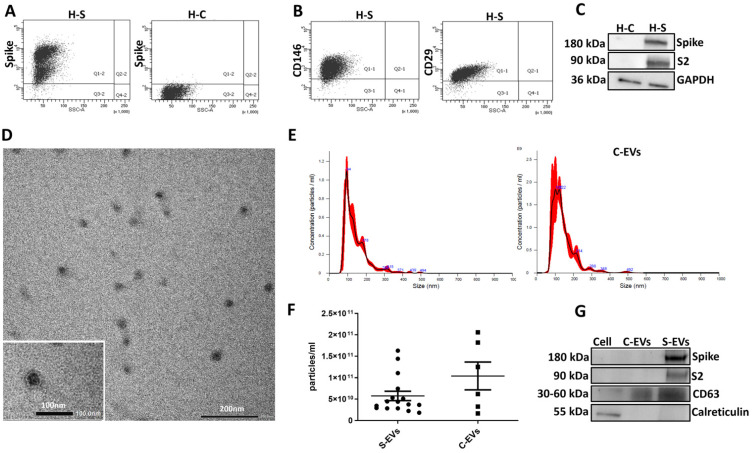
Characterization of H-S and EVs with validation of spike presence. (**A**) Representative flow cytometry analysis of spike protein in H-S and H-C. (**B**) Representative flow cytometry analysis of H-S showing positive expression of CD146 and CD29. (**C**) Representative western blot images of both full-length spike protein and S2 subunit (spike and S2) in H-S and H-C. GAPDH was used as an endogenous loading reference. (**D**) Representative micrograph of transmission electron microscopy of S-EVs (Scale bar: 200 nm; insert: 100 nm). (**E)** Representative nanoparticle tracking analysis of EVs from H-S cells (S-EVs) and from H-C cells (C-EVs) showing EV size distribution. (**F**) The graph shows EV sample quantifications. (**G**) Western blot images of spike subunits in S-EVs and C-EVs. CD63 was used as an exosomal marker and calreticulin as a negative EV marker.

**Figure 2 cells-11-00146-f002:**
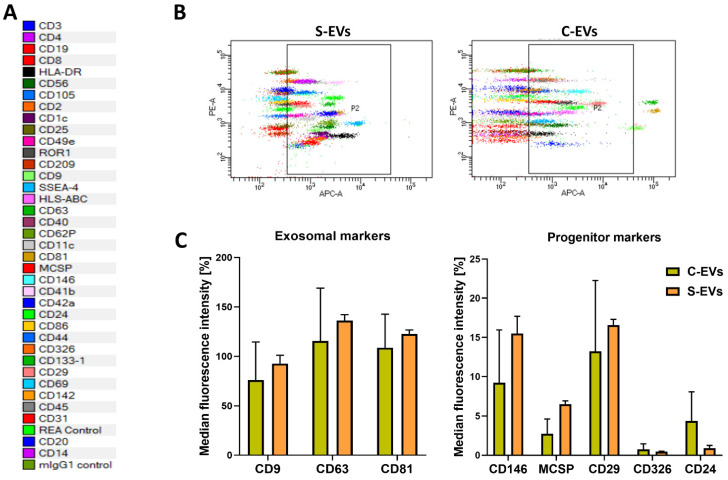
Characterization of S-EVs. (**A**) List showing the 39 antibodies used in the assay and their respective colors in dot plots. (**B**) MACSPlex representative dot plots showing the S-EVs and C-EVs distribution of allophycocyanin (APC)-stained bead populations. (**C**) Quantification of the median APC fluorescence for each bead population after background correction, clustered in exosomal and progenitor markers. The progenitor markers were normalized to median fluorescence intensity of exosomal markers. Data is expressed as the average of three independent experiments ± SD.

**Figure 3 cells-11-00146-f003:**
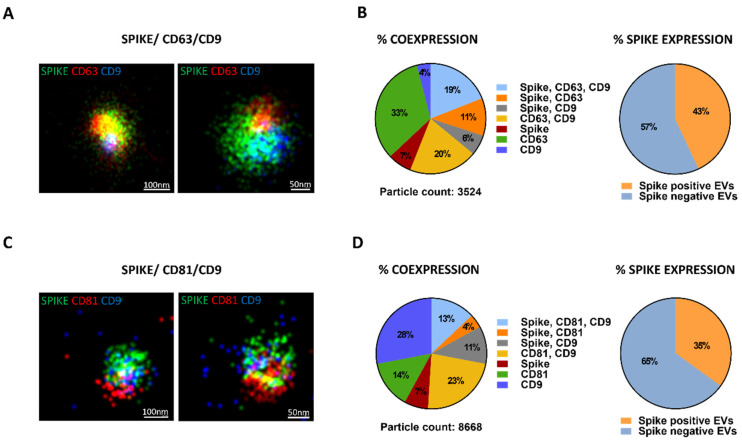
Super-resolution microscopy analysis of EVs isolated from H-S. (**A**) Super-resolution microscopy micrographs showing the pattern distribution of spike in green, CD63 in red and CD9 in blue (Spike/CD63/CD9) for S-EVs. (**B**) The percentage of EVs in triple, double or single positivity for spike, CD63, CD9 markers (% coexpression) and the total percentage of EVs positive or negative for spike protein (% spike expression) was reported. (**C**) Super-resolution microscopy micrographs showing the pattern distribution of spike in green, CD81 in red and CD9 in blue (Spike/CD81/CD9) for S-EVs. (**D**) The percentage of EVs in triple, double or single positivity for spike, CD81, CD9 markers (% coexpression) and the total percentage of EVs positive or negative for spike protein (% spike expression) was reported.

**Figure 4 cells-11-00146-f004:**
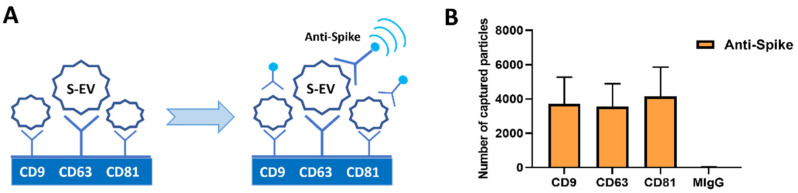
ExoView analysis of S-EVs. (**A**) Schematic representation of S-EVs detection process for ExoView technique. (**B**) Number of S-EVs captured on CD9, CD63, CD81 or MIgG spots fluorescently labeled by anti-spike ab in APC obtained by ExoView analysis. The graph shows the average of three independent experiments ± SD.

**Figure 5 cells-11-00146-f005:**
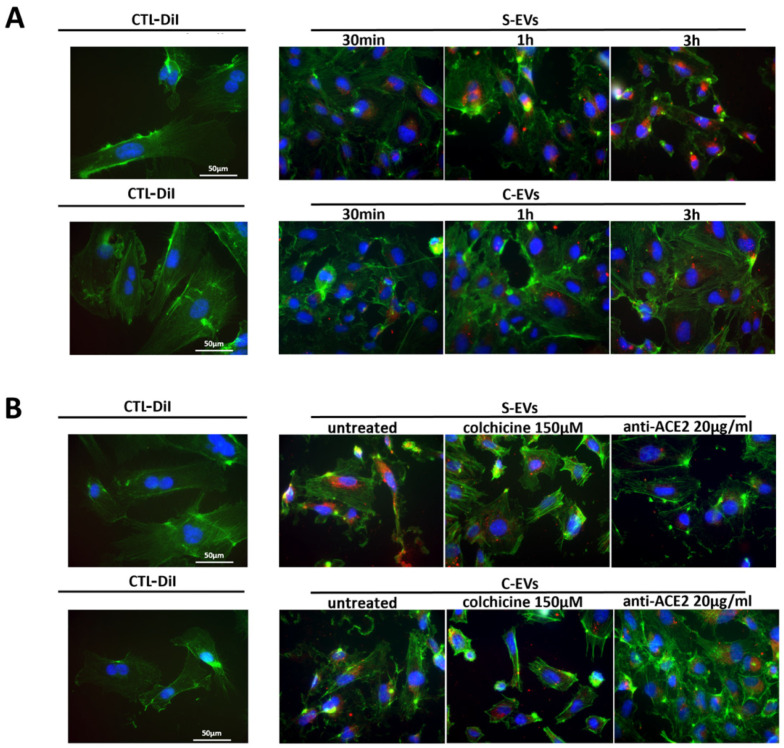
S-EV uptake and its modulation. (**A**) Representative immunofluorescence micrograph of S-EV or C-EV uptake by HUVEC after 30 min, 1 h, or 3 h with respect to the control (CTL-DiI), prepared with DiI control solution in the absence of EVs. (**B**) Representative immunofluorescence micrograph of S-EV or C-EV uptake modulation by HUVEC with colchicine or anti-ACE2 blocking antibody with respect to the uptake without treatments (untreated) or to control (CTL-DiI), prepared with DiI control solution in the absence of EVs. Cells were stained with FITC-phalloidin (green), nucleus-stained with DAPI (blue), EVs were labeled with DiI (red); magnification ×40.

**Figure 6 cells-11-00146-f006:**
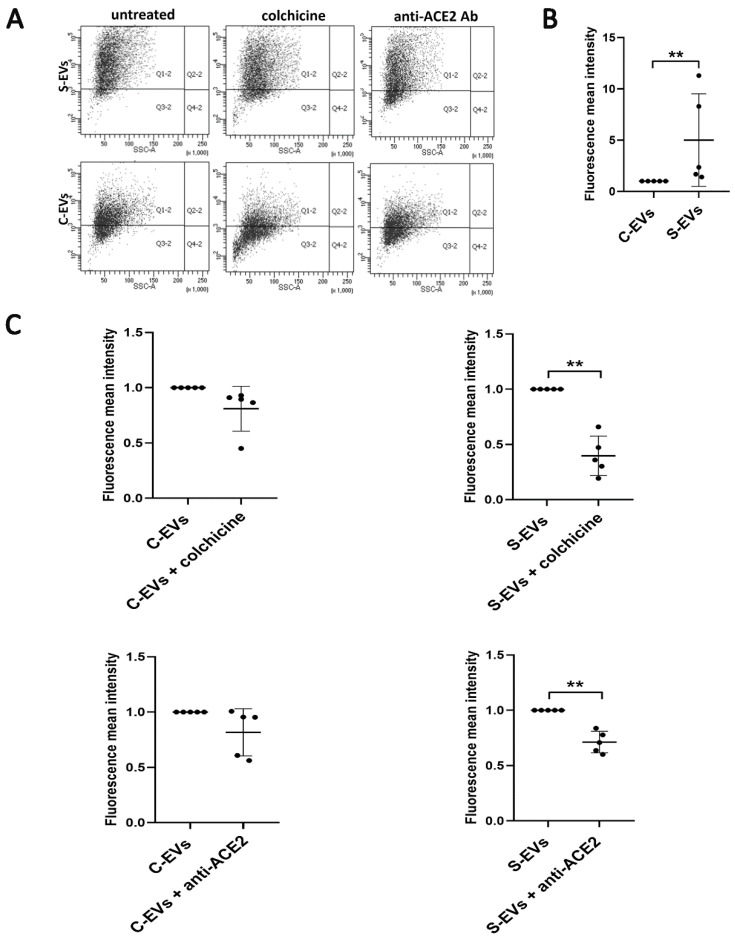
Effect of colchicine and anti-ACE2 blocking antibody on the EV uptake by HUVEC. (**A**) Representative flow cytometry dot plots of S-EV or C-EV uptake without treatments (untreated) or with colchicine 150 µM (S/C-EVs colchicine) or anti-ACE2 20 µg/mL (S/C-EVs anti-ACE2 Ab). (**B**) Fluorescence mean intensity of all the positive events obtained by cytofluorometric analysis. Data were normalized to the respective uptake control (S-EVs or C-EVs), set as one, used as a reference sample for each experiment. For the comparison, C-EVs vs. S-EVs data were normalized to the C-EVs. The unpaired *t*-test was performed after the normalization for C-EV vs S-EV uptake with ** *p* < 0.01. (**C**) Fluorescence mean intensity of all the positive events obtained by the cytofluorometric analysis. Data were normalized to the respective uptake control (S-EVs or C-EVs), set as one, used as a reference sample for each experiment. For the comparison EVs vs. EVs + colchicine or EVs + anti-ACE2, data were normalized to the EVs. The paired -*t*-test was performed after the normalization for EVs vs. EVs + colchicine or EVs + anti-ACE2 with ** *p* < 0.01. The graphs show the average of at least five independent experiments ± SD.

**Figure 7 cells-11-00146-f007:**
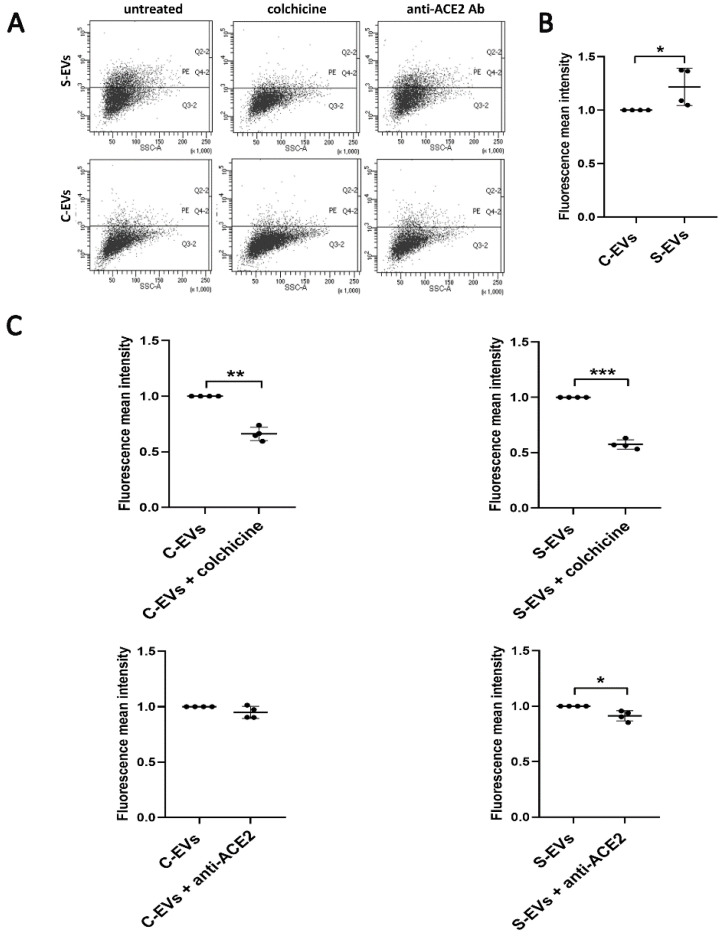
Effect of colchicine and anti-ACE2 blocking antibody on the EV uptake by the bronchial epithelial cell line 16HBE14o-. (**A**) Representative flow cytometry dot plots of S-EV or C-EV uptake without treatments (untreated) or with colchicine 150 µM (S/C-EVs colchicine) or anti-ACE2 20 µg/mL (S/C-EVs anti-ACE2 Ab). (**B**) Fluorescence mean intensity of all positive events obtained by cytofluorometric analysis. Data were normalized to the respective uptake control (S-EVs or C-EVs), set as one, used as a reference sample for each experiment. For the comparison C-EVs vs. S-EVs, data were normalized to the C-EVs. The unpaired *t*-test was performed after the normalization for C-EV vs. S-EV uptake with * *p* < 0.05. (**C**) Fluorescence mean intensity of all positive events obtained by cytofluorometric analysis. Data were normalized to the respective uptake control (S-EVs or C-EVs), set as one, used as a reference sample for each experiment. For the comparison EVs vs. EVs + colchicine or EVs + anti-ACE2, data were normalized to the EVs. The paired *t*-test was performed after the normalization for untreated EVs vs. EVs + treatments with * *p* < 0.05, ** *p* < 0.01 or *** *p* < 0.001. The graphs show the average of at least four independent experiments ± SD.

## Data Availability

Not applicable.

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
