# Peer review of "Generation of Spike-Extracellular Vesicles (S-EVs) as a Tool to Mimic SARS-CoV-2 Interaction with Host Cells"

_cells, 2022, doi:10.3390/cells11010146_

Round 1

Reviewer 1 Report

The study by Verta et al. aims at characterizing VLP that display the spike of SARS-CoV-2 at their surface. It is mostly a descriptive study that should be improved to reach the publication level.

Major concerns:

The H-S cell line has been commercially obtained but it would be necessary to describe how the cell line has been established.

Lane 44: the spike is mainly cleaved in the producer cells by furin and not after the binding. S is mainly found as a cleaved protein in EV or VLP.

The CD146 and CD29 staining in Fig. 1 don’t bring anything to the study.

Fig. 3B is not clear at all.

In the discussion, the following comment starting lane 323 is not correct. If you use an antibody directed against S2 you will detect S2 and S, but it does not say that you don’t have S1. If you would have used an antibody against S1, you would have detected S and S1.

Minor concerns:

Lane 50: results should be replaced by is.

Lane 55: extracellular vesicles should be replaced by EV.

Lane 56: their biogenesis.

Lane 194: spike should be replaced by S2.

Lane 243: CD63 and CD9 is written twice.

Lane 310: characterized.

Author Response

We thank the reviewer for the helpful suggestions, and we revised the manuscript accordingly.

Major concerns:

-The H-S cell line has been commercially obtained but it would be necessary to describe how the cell line has been established.

Response: We apologize for the lack of information about the H-S cells. Now, the details are reported in the Material and Methods, section 2.1 lanes 102-111

-Lane 44: the spike is mainly cleaved in the producer cells by furin and not after the binding. S is mainly found as a cleaved protein in EV or VLP.

Response: We thank the reviewer for the clarification; we deleted the wrong sentence. Lane 50-51

-The CD146 and CD29 staining in Fig. 1 don’t bring anything to the study.

Response. The experiments in figure 1B give evidence about the stable state of cells, that preserve the typical expression of progenitor markers (Togarrati P.P. et al,2018 and Hörl, S. et al, 2017), after the spike vector transfection. It’s important information to start the EVs isolation from the H-S cells. Therefore, we maintained the data in Figure 1. However, they can be moved to the Supplemental figures if required.

Fig. 3B is not clear at all.

Response. We apologize and revised the Figure to make it clearer and easier to understand. The actual Figure 3 is on page 8.

Minor concerns:

Lane 50: results should be replaced by is. Thanks, we replaced results by “is”, lane 58

Lane 55: extracellular vesicles should be replaced by EV. Thanks, we changed using the correct abbreviation “EV”, lane 63

Lane 56: their biogenesis. Thanks, we added “their”, lane 64

Lane 194: spike should be replaced by S2. Thanks, we replaced spike by “S2”, lane 224

Lane 243: CD63 and CD9 is written twice. Thanks, we removed the repetition, lane 276

Lane 310: characterized. Thanks, we corrected, lane 371

Reviewer 2 Report

There is no benefit to using EV to deliver SP to other cells. Why not directly use SP. 

Author Response

As the aim of the study was to use Spike-EV internalization as a tool to assess the effect of different drugs on viral entry, SP alone cannot be used. We agree that SP alone can also be of benefit, related to the evaluation of ACE-2 binding.

Reviewer 3 Report

 Generation of Spike- Extracellular Vesicles (S-EVs) as a Tool to Mimic SARS-CoV-2 Interaction with Host Cells

Roberta Verta et al.

Summary

In brief, this work demonstrates the feasibility of producing extracellular vesicles (EVs) expressing SARS-CoV-2 spike protein, and further documents their uptake by human umibilical endothelial cells. The investigators employ a number of state-of-the-art techniques to prove and validate their results and conclusions. Overall, the investigation reinforces the idea that EVs can be used to study SARs-CoV-2-endothelial cell interactions and to develop potential therapeutic strategies.

Major Comments

  1. The data in Figure 6 are not so convincing. First, it would be helpful to display individual data points (scatter graphs) with superimposed mean and SE/SD, or box and whiskers plots. It is difficult to appreciate the variabilty of the data, but it must be quite large in several instances (e.g., panel A, S-EVs). Second, it appears as if the uptake of S-EVs and C-EVs may have been similarly blocked by the anti-ACE2 antibody. The qualitative reduction was by 30% or so in both cases, although only the latter was significant. These results need clarification and perhaps by completing additional experiments due to the apparently large variability. (Indeed, n=3 is inadequate in general as a representative sample of any population.) As it stands now, it is possible that the EVs whether C or expressing S protein are gaining access to the cells largely through S protein-independent mechanisms. This interpretation would seem to negate the usefulness of S-EVs in the study of SARs-CoV2 interaction with endothelial cells.
  1. The invesigators utilized HUVEC cells which are of fetal origin. It would be important for the authors to corroborate some of their most important findings (e.g., Fig. 6) using endothelial cells derived from adults, e.g., aortic, coronary, microvascular, also obtained from ATCC or Lonza.

Minor Comments

  1. Although generally well-written, there are a number of issues where English usage could be improved. Perhaps the authors could ask a colleague fluent in English (or a native English speaker) to edit the manuscript.

Author Response

In brief, this work demonstrates the feasibility of producing extracellular vesicles (EVs) expressing SARS-CoV-2 spike protein, and further documents their uptake by human umbilical endothelial cells. The investigators employ a number of state-of-the-art techniques to prove and validate their results and conclusions. Overall, the investigation reinforces the idea that EVs can be used to study SARs-CoV-2-endothelial cell interactions and to develop potential therapeutic strategies.

We thank the reviewers for the positive comments and for the careful evaluation of our manuscript.

Major Comments

- The data in Figure 6 are not so convincing. First, it would be helpful to display individual data points (scatter graphs) with superimposed mean and SE/SD, or box and whiskers plots. It is difficult to appreciate the variability of the data, but it must be quite large in several instances (e.g., panel A, S-EVs).

Response. According to the request, we displayed the single data point by scatter graphs, with mean and SD in Figure 6 as well as in the new generated Figure 7.

-Second, it appears as if the uptake of S-EVs and C-EVs may have been similarly blocked by the anti-ACE2 antibody. The qualitative reduction was by 30% or so in both cases, although only the latter was significant. These results need clarification and perhaps by completing additional experiments due to the apparently large variability. (Indeed, n=3 is inadequate in general as a representative sample of any population.) As it stands now, it is possible that the EVs whether C or expressing S protein are gaining access to the cells largely through S protein-independent mechanisms. This interpretation would seem to negate the usefulness of S-EVs in the study of SARs-CoV2 interaction with endothelial cells.

Response. We appreciate the suggestion of the reviewer to increase the number of replicates in our experiments. We now generated a total of five experiments for each condition. It is now clearer that a high decrease of internalization by anti-ACE2 blocking antibody is observed in the S-EVs, where it reaches the statistical significance, whereas in the control a lower and non-significant effect is present. See Figure 6 and figure legend.

-The investigators utilized HUVEC cells which are of fetal origin. It would be important for the authors to corroborate some of their most important findings (e.g., Fig. 6) using endothelial cells derived from adults, e.g., aortic, coronary, microvascular, also obtained from ATCC or Lonza.

Response. We thank the reviewer for the proposal to investigate the EV interaction with adult endothelial cells. According to the comment, we tested the ACE2 expression on human microvascular endothelial cells (HMEC) from Lonza. Unfortunately, the spike receptor expression was very low (4,5%) compared to the HUVEC and unusable for the S-EV uptake experiments. See Figure uploaded.

Analysis of ACE2 expression on HMEC. Representative flow cytometry analysis of Human microvascular endothelial cells (HMEC) showing the negative staining of a control isotype (ISO-FITC) and the positive expression of ACE2 (ACE2-FITC).

Therefore, we decided to focus on human bronchial epithelial cells (16HBE14o-), target site for SARS-CoV-2 infection (Liao, Y. et al., 2020; Blume, C. et al., 2021 and Kam, Y.W. 2009). The ACE2 expression on 16HBE14o- was comparable to the HUVEC. The data was added in the new supplementary figure 1. On these cells, we performed parallel experiments on EV uptake and treatment with colchicine or anti-ACE2 blocking antibody. The EV entrance reduction by anti-ACE2 treatment was significant only for the S-EVs a not for the C-EVs. Moreover, a significant reduction for both S-EV and C-EV uptake was observed by colchicine (Figure 7). These results confirmed the indirect colchicine effect previously observed on the HUVEC by FACS and IF analysis (Figure 7). All these data support the finding before obtained using HUVEC.

See Figure 7, Materials and Methods, cytofluorimetric analysis section, lane 204 and lane 214; Results, Modulation of S-EV uptake section, lane 349-357; and Discussion lanes 413-421.

Reviewer 4 Report

The proposed paper highlight an original way to study COVID - and widely virus infection - using small EVs holding spikes. However, some points for the EV part need to be more investigated and revised:

  • MISEV (Lötvall et al., 2014 JEV; Théry et al., 2018 JEV) recommendations have to be considered to better characterize your small EVs, such as presence/absence of markers (CD9, 63, 81 OK but what else?) and at least a view of the EVs (TEM or AFM).
  • IF can also be improved to better see the EVs, a negative control without EVs to see the naive auto fluorescence would be appreciable to distinguish labeled EVs from alto fluorescent dots within the cytoplasm.

Author Response

We thank the reviewer for the useful comments and suggestions.

MISEV (Lötvall et al., 2014 JEV; Théry et al., 2018 JEV) recommendations have to be considered to better characterize your small EVs, such as presence/absence of markers (CD9, 63, 81 OK but what else?) and at least a view of the EVs (TEM or AFM).

Response. We thank the Reviewer for the suggestion as it allowed us to obtain a better and complete S-EV characterization. According to the request, in addition to the use of different techniques including Nanoparticle tracking analysis, Western blot, MACSPlex, ExoView and Super-resolution microscope analysis, we also now performed the Transmission electron microscopy (TEM) evaluation. By TEM, we showed the EV shape and size. A representative micrograph of transmission electron microscopy is added in Figure 1D.  

See Figure 1D, materials and methods lanes 160-166,  results lanes 248-254.

IF can also be improved to better see the EVs, a negative control without EVs to see the naive autofluorescence would be appreciable to distinguish labeled EVs from alto fluorescent dots within the cytoplasm.

Response. To appreciate the labeled EVs in respect to the possible autofluorescence background, as requested, we used a negative control solution (CTL-DiI). CTL-DiI was obtained using the same protocol for the DiI-labeled EV in the absence of the EVs. The IF analysis did not show autofluorescence and allowed to appreciate the labeled EVs.

See micrographs in Figure 5, and material and methods section lanes 190-192.

Round 2

Reviewer 2 Report

Then please provide direct evidence that spike protein EV enter cells same as the virus.

Reviewer 3 Report

The authors nicely responded to the reviewer comments and suggestions.

Reviewer 4 Report

I would like to thank the authors to have revised the paper following my recommendations. All the modifications done make this paper acceptable in this revised version.